# Towards Mixed-Initiative Generation of Multi-Channel Sequential Structure

**Anna Huang**[1]**, Sherol Chen**[1]**, Mark J. Nelson**[2]**, Douglas Eck**[1]
[1]Google Brain, Mountain View, CA 94043, USA
[2]The MetaMakers Institute, Falmouth University, Cornwall, UK

`annahuang@google.com, sherol@google.com,`
`mjn@anadrome.org, deck@google.com`

## ABSTRACT

We argue for the benefit of designing deep generative models through a *mixed-initiative, co-creative* combination of deep learning algorithms and human specifications, focusing on multi-channel music composition. Sequence models have shown convincing results in domains such as summarization and translation; however, longer-term structure remains a major challenge. Given lengthy inputs and outputs, deep generative systems still lack reliable representations of beginnings, middles, and ends, which are standard aspects of creating content in domains such as music composition. This paper aims to contribute a framework for mixed-initiative generation approaches that let humans both supply and control some of these aspects in deep generative models for music, and present a case study of Counterpoint by Convolutional Neural Network (CoCoNet) (Huang et al., 2017).

## 1 INTRODUCTION

As generative machine learning models improve in their ability to imitate and transfer rich creative artifacts and styles, they become increasingly useful in creative pursuits. However, we often do not want the machine to drive the entire generative process. Humans in the loop can allow generation to capture subjective and context-dependent preferences, and in complex tasks can simplify the modeling problem. In addition, humans working with ML systems often *want* to meaningfully shape the result.

We focus on musical composition, which provides not only formally identified structures, but also a language that is easily represented both visually and numerically. It has a sequential structure, but in polyphonic music it is a multi-channel sequential structure with coordinated and synchronized timelines. We believe that insights from deep generative music models can carry over to other domains that have multi-channel sequential content, such as screen plays and video games. Thus, it is a suitable path for studying how this type of content can be effectively and constructively generated in a mixed-initiative manner.

## 2 MIXED-INITIATIVE DEEP GENERATIVE MODELS

We propose that a framework of *mixed-initiative co-creativity* (Horvitz, 1999; Liapis et al., 2016; Deterding et al., 2017) clarifies the design space for deep-learning generative systems interacting with humans, where each drives aspects of the creative process. *Mixed initiative* means designing interfaces where a human and an AI system can each "take the initiative" in making decisions. *Co-creative* means building generative systems where creative outputs are driven by meaningful creative input from both generative techniques and humans.

The large literature on mixed-initiative and co-creative systems can help us understand and design tradeoffs and possibilities when building generative ML systems that create together with humans. To choose just a few examples of useful concepts to borrow: human/machine co-creation can have disparate goals, ranging from a creative coach to a colleague, with UX and system-design impli-

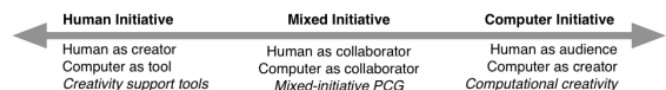

Figure 1: Spectrum of human and computer initiatives (Deterding et al., 2017)

cations (Lubart, 2005); the loaded term *autonomous* can be broken down into more precise technical meanings (Bradshaw et al., 2003); co-creation can be structured temporally (alternating) or by competency (task-divided) (Kantosalo & Toivonen, 2016); and interactive ML techniques such as imitation learning can help scale control of co-creative systems (Jacob & Magerko, 2015).

Here we focus on two aspects of this design space specifically for generating multi-channel musical structure. One is a simple axis of initiative, from mainly human to mainly computer initiative, illustrated in Figure 1. The other is a qualitative look at how the user makes decisions.

Some decisions are surface-level changes, such as modifying a few notes in a musical score or a small patch of pixels on an image, usually through direct manipulation. Others are high-level changes, such as changing the mode of a phrase from major to minor. High-level changes usually imply many surface-level changes, so require a way to map between the desired high-level change and the required low-level changes.

Many methods have been proposed to map from high-level to surface-level changes in generative music systems. An ML system's existing parameters can be directly exposed as knobs (Morris et al., 2008); latent spaces can be used through interpolation for composing transitions Roberts & Engel (2017) or as a palette for exploration (Roberts et al., 2018); a specification language can allow for configurable constraints on latent spaces (Engel et al., 2017); in a reinforcement learning setting, musical constraints can be used to formulate rewards to tune a generative system (Jaques et al., 2017); or musical examples can be used as templates whose high-level structure is extracted and used as optimization objectives for guiding the synthesis of a new piece (Lattner et al., 2016).[1]

Three primary factors for the system designer are: (1) which controls or parameters the user will specify to guide the generation process, (2) how these controls impact the generator's processes, and (3) the amount of overall control this gives the user over the produced content.

## 3 COCONET: PARTIAL SCORE COMPLETION

Machine learning models of music typically break up the task of composition into a chronological process, composing a piece of music in a single pass from beginning to end. On the contrary, human composers write music in a nonlinear fashion, scribbling motifs here and there, often revisiting choices previously made. This motivates the need to design models that are more flexible.

COCONET is a deep convolutional neural network trained to reconstruct partial scores (Huang et al., 2017), analogous to orderless NADE (Uria et al., 2014; 2016) . Musical scores are represented as multi-channel 2D binary matrices, encoding instruments, discretized time and pitch (see Figure 3 for an example). Once trained, the model provides direct access to all conditionals of the form $p(\mathbf{x}_i \mid \mathbf{x}_C)$ where $\mathbf{x}_C$ is a fragment of a musical score $\mathbf{x}$ and $i \notin C$ is in its complement.

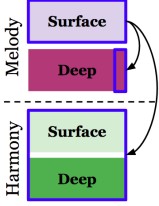

Figure 2: Showing change propagation across channels from the melodic surface to underlying harmonic structure.

Given a partial score from the user, COCONET can use these conditions to fill in the gaps. In practice, we use blocked Gibbs sampling which repeats the in-filling process to approximate the benefits of rewriting. Since COCONET supports general partial score completion, musicians can use it to accomplish a wide range of musical tasks, such as unconditioned

---

[1]The last method is parallel to earlier work in visual style transfer where examples are given for content and style (Gatys et al., 2016; Dumoulin et al., 2017).

generation, harmonization, transition, rewriting an existing piece by by removing and regenerating voice by voice.

### 3.1 MIXED-INITIATIVE CHANGE PROPAGATION

As opposed to a two stage process where the human first composes some parts of the score, and the model completes the rest, the human and the model can engage in interleaved turn-taking during the creative process. We give an example of how machines can help users explore changes by quickly prototyping the potential impact of their changes. The musician composes a melody to outline the beginning, middle and end of a piece, closing with a downward contour. Here we substitute the melody with "Ode to joy" so that it is recognizable. The musician asks "what if" she moved the last two notes up to add a twist, would that break the closure? Even though the user only changed the musical surface of the melody, it could imply deeper harmonic changes. Seeing the new melody, COCONET regenerates the other chan-

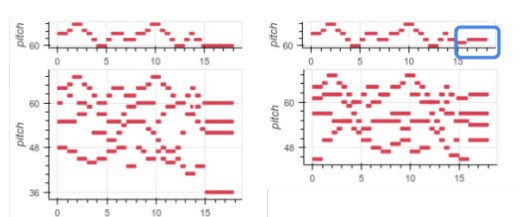

Figure 3: COCONET helps the user imagine how a piece would sound if she changed the last two notes of the melody to the ones bracketed in blue. Left: the original melody on top, and COCONET's harmonization on the bottom. Right: the user's modified melody, and COCONET's new harmonization.

nels where not only the pitches are changed but also the key. However, COCONET was able to preserve the closure and make the "wrong" notes sound right. But if surprise was the intention of the user, then the turn-taking continues. The user can implicitly control the "abruptness" of the new changes by limiting how far back in the piece the machine can modify. Figure 3 shows the pianorolls of the original and modified melody and COCONET's harmonizations [2]. Figure 2 shows a schematic of the user's changes propagates across channels from surface to deep structure.

## 4 CONCLUSION

Deep generative systems' potential can be maximized if humans are integrated into the generative loop—at least when they want to be. This will allow humans to create in new ways that are enabled by contemporary machine learning. In addition, humans do have many advantages in artistic domains, such as more direct access to their own subjective preferences and context, whose modeling can be minimized or bypassed if the user can directly control them.

We proposed *mixed-initiative co-creativity* as a framework for thinking about the design options and tradeoffs when building generative ML systems that create together with humans. By adapting these existing HCI and AI concepts to the new possibilities of deep generative systems, we can more purposefully understand what kind of human/machine interaction we are looking for, and design purposely for it. We specifically focus here on multi-channel sequential music generation in the deep generative system COCONET, which we believe is a domain well suited to investigating such mixed-initiative generative systems.

In future work we would like to extend this approach to complex sequential domains other than music, such as stories. For example, RoleModel is a constraint-satisfaction based story generator that, like COCONET, takes specifications from the story author and infers from a model how to enhance and complete the composition (Chen et al., 2010). This is not a machine-learning approach, instead breaking stories down into grammars and rulesets that the author can take the initiative to recombine as desired, while also giving the computer initiative in completing outcomes. In expanding mixed-initiative deep generative models beyond music, we hope to borrow from these practices to create similar modes of engagement between human and computer.

---

[2]Samples of COCONET's harmonization of the original "Ode to joy" melody and the melody after the modification on the last two notes can be heard at `https://coconets.github.io/`

ACKNOWLEDGMENTS

We would like to thank Natasha Jaques for her contributions to this project. Special thanks to Stefaan De Rycke for adding the final twist to "Ode to Joy".

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
