# OpenReview forum: "Towards Mixed-initiative generation of multi-channel sequential structure"
_ICLR.cc/2018/Workshop — Accept_

### Official Review · AnonReviewer4 · 2018-02-26
**one of the official reviews**

**Rating:** 8
**Confidence:** 4

**Review:**

This workshop contribution examines a mixed-initiative approach to human-computer interaction via turns between human and computer, with application to music composition.  In particular, the computational creativity aspect is accomplished via CoCoNet.

This is a very important topic of inquiry because computational creativity is and will continue to be a critical branch of artificial intelligence for society, whether it is in creating novel music, text, or molecules.  The human interaction is just as critical as the authors excellently justify.  These topics should be discussed at ICLR to help ensure the field moves towards creativity research.

The general concept of mixed-initiative co-creation is not new (and it is not claimed to be either).  The novel contribution in this work is expanding on CoCoNet for this regime.

The paper is well written for the amount of allocated space.

Some form of evaluation of the results would be appreciated.  It might also be good to briefly discuss other hierarchical machine learning approaches for music, e.g. https://openreview.net/pdf?id=ryhqQFKgl.

---

### Official Review · AnonReviewer3 · 2018-03-09
**A good framework with a good example for human-machine collaborative creativity**

**Rating:** 7
**Confidence:** 4

**Review:**

This position paper proposes thinking about the task of human-machine collaboration in creative endeavors along the axes of initiative and the types of controls the human is given in the interaction.  It provides a concrete example of such a system in CoCoNet and the ways that it can be used for tasks in different locations on these axes.  CoCoNet was recently introduced at ISMIR 2017 and is a model inspired by NADE, but over a piano roll representation of a musical score.  Instead of generating music sequentially, it provides a mechanism by which a creation or parts of it can be iteratively refined.  The user-control axis in this case is the types of modifications the user makes to this shared score representation, to the melody, harmony, or leaving blanks to be filled in by the model.  The initiative axis is controlled by when and how the human modification of the score is interleaved with the model's new proposals.

One issue that was not clear in the paper is the novelty of this perspective.  It states that there is a "large literature on mixed-initiative and co-creative systems" so it seems that th novelty is in applying this to deep generative models of music, but this could perhaps be stated more explicitly.

Overall, this is an interesting perspective with a compelling concrete example that can help conceptualize these kinds of collaborations between humans and generative models.

---

### Official Review · AnonReviewer1 · 2018-03-10
**Sensible objective, but technical contents are thin**

**Rating:** 3
**Confidence:** 4

**Review:**

This paper proposes an architecture of "humans-in-the-loop" for generating
multi-channel structures such as music.
For this purpose, the authors propose COCONET that executes automatic partial
score completion as an orderless NADE, and works like a Gibbs sampler.

I appreciate the objective and architectures of the proposed method. Indeed,
generation of such multi-channel structure is not well studied and will be an
important research area in the near future. However, this paper lacks neither
technical descriptions of the proposed COCONET nor comparison with previous
attempts for music generation. Lacking of description of technical difference
from previous research, I cannot see this paper to be accepted as ICLR even as
a workshop paper.

---

### Decision · Program_Chairs · 2018-03-20
**ICLR 2018 Workshop Acceptance Decision**

**Decision:**

Accept

**Comment:**

Congratulations, your paper was accepted to the ICLR workshop.